# Optimum Fermentation Conditions for Bovine Lactoferricin-Lactoferrampin-Encoding *LimosiLactobacillus reuteri* and Regulation of Intestinal Inflammation

**DOI:** 10.3390/foods12224068

**Published:** 2023-11-09

**Authors:** Weichun Xie, Xueying Wang, Jiyao Cai, Huitao Bai, Yilan Shao, Zhuoran Li, Limeng Cai, Senhao Zhang, Jiaxuan Li, Wen Cui, Yanping Jiang, Lijie Tang

**Affiliations:** 1College of Veterinary Medicine, Northeast Agricultural University, Harbin 150030, China; xieweichun_neau@163.com (W.X.); tycoon28644@163.com (X.W.); caijiyao0723@163.com (J.C.); bht17077876222@163.com (H.B.); shaoyilan@neau.edu.cn (Y.S.); lizhuoran1009@163.com (Z.L.); clmyoyo@163.com (L.C.); zhang_sen_hao@163.com (S.Z.); lijiaxuan.1993@163.com (J.L.); cuiwen@neau.edu.cn (W.C.); 2Heilongjiang Key Laboratory for Animal Disease Control and Pharmaceutical Development, Northeast Agricultural University, Harbin 150030, China

**Keywords:** lactoferricin-lactoferrampin (LFCA), *LimosiLactobacillus reuteri*, protein expression, high-density fermentation, intestinal inflammation

## Abstract

The multifunctional antibacterial peptide lactoferricin-lactoferrampin (LFCA) is derived from bovine lactoferrin. Optimization of the fermentation process should be studied since different microorganisms have their own favorable conditions and processes for growth and the production of metabolites. In this study, the culture conditions of a recombinant strain, pPG-LFCA-E/LR-CO21 (LR-LFCA), expressing LFCA was optimized, utilizing the high-density fermentation process to augment the biomass of *LimosiLactobacillus reuteri* and the expression of LFCA. Furthermore, an assessment of the protective effect of LR-LFCA on intestinal inflammation induced by lipopolysaccharide (LPS) was conducted to evaluate the impact of LR-LFCA on the disease resistance of piglets. The findings of this study indicate that LR-LFCA fermentation conditions optimally include 2% inoculation volume, 36.5 °C fermentation temperature, 9% dissolved oxygen concentration, 200 revolutions/minute stirring speed, pH 6, 10 mL/h glucose flow, and 50% glucose concentration. The inclusion of fermented LR-LFCA in the diet resulted in an elevation of immunoglobulin levels, significant upregulation of tight junction proteins ZO-1 and occludin, reinforcement of the intestinal barrier function, and significant amelioration of the aberrant alterations in blood physiological parameters induced by LPS. These results offer a theoretical framework for the implementation of this micro-ecological preparation in the field of piglet production to enhance intestinal well-being.

## 1. Introduction

Lactoferrin (LF) is a protein that binds to iron and is a member of the transferrin family. It is abundant in mammalian milk and exhibits broad-spectrum antimicrobial properties, as well as various other functions, including anti-inflammatory, antioxidant, and anti-tumor effects [1]. Lfcin (the fragment lactoferricin is generated during digestion of lactoferrin by pepsin [2]) is a cyclic peptide derived from the 17–41 amino acid residues of the bovine lactoferrin peptide chain, formed by two cysteine residues that create a disulfide bond [3]. Furthermore, the N1 region of bovine lactoferrin contains a peptide segment known as LFampin (a second antimicrobial LF synthetic peptide was obtained and designated lactoferrampin by van der Kraan et al. [4]), which shares a similar structure with LFcin and is composed of residues 268–284 [5]. LFampin possesses a significant quantity of cationic charges and demonstrates pronounced hydrophobicity, enabling its adhesion to the exterior of bacterial cell membranes and consequent disruption, ultimately resulting in antibacterial efficacy [6]. Lactoferrin peptide exhibits anti-inflammatory and immune-regulatory properties [7]. Lfcin, a type of antimicrobial protein, demonstrates a robust binding affinity for lipopolysaccharide (LPS), thereby mitigating its toxicity through direct neutralization [8].

The intestine serves as a crucial location for the digestion and absorption of nutrients in animals [9]. Lactoferrin peptide, functioning as an antimicrobial agent, plays a significant role in enhancing intestinal health by leveraging its antibacterial and immune regulatory properties [10]. Lactic acid bacteria (LAB) have been acknowledged as promising contenders for the advancement of innovative and reliable heterologous protein production and delivery systems [11]. *LimosiLactobacillus reuteri* exhibits resistance to gastric acid and bile salts, along with a certain survival rate within the intestinal tract [12]. The *L. reuteri* strain utilized in this investigation was isolated from the intestines of piglets, and has been shown to modulate host protective immune responses [13,14]. The ability to synthesize specific proteins and the upper limit of production capacity that can be attained are contingent upon the recombinant construction of engineered bacteria. However, the extent to which this potential can be actualized is reliant upon the advancement of high-density fermentation technology [15]. Optimization of fermentation primarily enhances production efficiency and diminishes production expenses by optimizing fermentation conditions (e.g., pH, temperature, stirring speed) and appropriate culture medium components (e.g., carbon and nitrogen sources).

Through the transportation and metabolism of extracellular nutrients, the regulation and control of precursor and intermediate accumulation can be achieved at various levels, ultimately leading to the attainment of high yields of specific metabolites [16]. In the pursuit of optimizing the culture medium, it is imperative to satisfy the optimal growth prerequisites of microorganisms to attain the maximal metabolite yield [17]. Optimization of fermentation conditions typically involves manipulating of multiple factors, including pH, temperature, stirring speed, and inoculation amount [18]. Recent research has demonstrated that microfluidic devices can offer precise regulation of these fermentation parameters, so as to increase production of targeted metabolites [19]. Response surface methodology (RSM) is a statistical technique utilized to simulate the impact of individual factors and their interactions on the model, as well as to identify optimal formula through equation modeling. RSM models are simple and efficient in forecasting the optimization outcomes of diverse metabolite productions [20].

LPS has been extensively employed to establish animal models of inflammation or bacterial infection [21]. In a porcine model of inflammation, intraperitoneal injection of high doses of LPS lead to acute inflammation, particularly intestinal inflammatory injury, and can result in multiple organ failure in severe cases [22]. LPS activates the immune system and accurately simulate the intestinal inflammatory response induced by Gram-negative bacteria [23]. This model has the distinct advantages of excellent repeatability and a high success rate, which is a crucial method for investigating acute intestinal inflammation model initiated through infection by pathogens [24].

In this study, our goal was to refine the fermentation process and culture conditions of recombinant *LimosiLactobacillus reuteri* that expresses lactoferrin peptides to augment the biomass of *L. reuteri* and the expression level of lactoferrin peptides. Subsequently, the fermented recombinant bacteria will be incorporated into the diet of piglets, and animal experiments will be conducted to corroborate the immune protective impact of recombinant bacteria against LPS-induced intestinal inflammation. This will furnish compelling evidence for the advancement and large-scale production of innovative microbial preparations, and investigate the influence of the research product on immune function and intestinal health via animal experimentation.

## 2. Materials and Methods

### 2.1. Bacterial Strains and Growth Conditions

*L. reuteri* CO21 was isolated from the intestinal contents of pigs and deposited at the China Center for Type Culture Collection (CCTCC) under accession number CCTCC M 2019601. The 16S rDNA sequence of this strain was deposited in GenBank (accession No. MK920155). The GenBank accession number for LFCA is WGT80058.1. LR-LFCA and *L. reuteri* CO21 transformed with a pPG612-EGFP plasmid (LR-CON) were constructed and preserved in our laboratory (Laboratory of Microbiology and Immunology, College of Veterinary Medicine, Northeast Agricultural University). Genetically modified strains of *L. reuteri* (LR-LFCA and LR-CON) were grown in MRS broth (De Man, Rogosa and Sharpe, Oxoid, Hampshire, UK) containing 10 µg/mL chloramphenicol at 37 °C for 16 h. *Staphylococcus aureus* (*S. aureus*, CVCC43300) and *Escherichia coli* (*E. coli*, CVCC10141) were cultured in LB broth (Oxoid, Hampshire, UK) for 12 h to reach 10^8^ CFU/mL in a constant temperature incubator (HPS-250, Harbin, China). The bacterial growth conditions have been described previously [13].

### 2.2. Determination of Growth Curve and Fermentation Culture of Recombinant LimosiLactobacillus reuteri

The recombinant *Lactobacillus* strain LR-LFCA was retrieved from −80 °C storage and subsequently streaked onto MRS solid plates. Following incubation at 37 °C until single colonies emerged, a single colony was selected and inoculated into a shaking flask containing 30 mL of MRS liquid medium. The flask was then incubated at 37 °C for 12 h to generate seed solution for fermentation. The bacterial growth conditions have been described previously [25,26]. LR-LFCA seed solution was subsequently inoculated at a 1% ratio into fermenter (DASGIP Eppendorf, Jülich, Germany) containing 3 L of fermentation medium. The experimental conditions were standardized with a stirring speed of 100 r/min, no air supply, and a temperature of 35 °C, with pH controlled at 7 using ammonia solution (5% *w*/*v*). The parameters screened are inoculum size, aeration rate, stirring speed, temperature, pH, and dissolved oxygen value, while maintaining other parameters constant. Sampling was performed at suitable intervals, and subsequent analysis was conducted upon completion of the experiment. In order to ascertain the growth conditions of LR-LFCA, LR-LFCA in the logarithmic growth phase were inoculated at a 1% ratio into MRS liquid medium and incubated statically at 37 °C for 24 h. The OD_600_ value of culture was measured every 2 h, with sterile MRS liquid medium used as negative control. The experiment was replicated three times, and the average value was calculated.

### 2.3. Western Blot Expression Identification and Quantitative Determination of LFCA Concentration

The experimental design is similar to our previous reports [13]. To obtain the supernatant, the fermentation liquid underwent centrifugation at 5000 r/min for 10 min. Bacterial cells were lysed through the use of 1% lysozyme and incubated at 37 °C for 2 h. The lysate underwent sonication and centrifugation at 12,000 r/min for 10 min, resulting in the protein sample from the recombinant bacteria precipitate. The same methodology was employed to prepare the LR-CON protein sample, which served as the negative control. The chemically synthesized bovine lactoferrin peptide standard was diluted in PBS to achieve a gradient of concentrations ranging from 1 to 100 μg/mL. The diluted peptide standard was immobilized onto a 96-well enzyme-labeled plate and subsequently blocked. The primary antibody, anti-bovine lactoferrin monoclonal antibody (diluted 1:400, prepared by our laboratory), and the secondary antibody, HRP-conjugated goat anti-mouse IgG (diluted 1:5000, Abcam, Cambridge, UK), were employed. The absorbance at 450 nm was measured using an ELISA reader (BioRad-680, Bio-Rad Ltd., Tokyo, Japan) to generate the standard curve. OD_450_ values of samples were utilized to determine peptide content in bacteria precipitate and supernatant.

Culture supernatants were collected 48 h and concentrated 100-fold by TCA as the method previously described [27]. Total protein was denatured by boiling for 10 min and electrophoresed on 15% Tris-Tricine SDS-PAGE gels. Primary antibodies: anti-MYC tag (1:1000 dilution, ABclonal, Cambridge, MA, USA). Secondary antibodies: goat-anti-mouse IgG HRP-conjugated (1:5000 dilution, Zhongshan Golden Bridge, Beijing, China).

### 2.4. Single-Factor Screening of Inoculation Amount, Culture Temperature, Stirring Speed, Dissolved Oxygen, and ph

The number of viable bacteria in fermentation seed liquid was 10^7^ CFU/mL. To ascertain the optimal inoculation size, the fermentation solution was inoculated with 10^8^ CFU/L (1%), 2 × 10^8^ CFU/L (2%), 3 × 10^8^ CFU/L (3%), 4 × 10^8^ CFU/L (4%), and 5 × 10^8^ CFU/L (5%), respectively. Fermentation was carried out in a fermenter with the parameters set at 100 r/min, pH 7, dissolved oxygen at 0%, and the temperature at 37 °C. Samples for OD_600_ determination and LFCA protein analysis were taken at regular time intervals. The constant temperature fermentation set as 33 °C, 35 °C, 37 °C, 39 °C, and 41 °C, while inoculum size 1%, other parameters were left unchanged, for determination of the temperature optimum. Similarly, for determination of the optimum stirring speed, dissolved oxygen, and culture medium pH, different culture conditions (stirring speed at 100, 200, 300, 400, and 500 r/min; dissolved oxygen at 0, 5, 10, 15, and 20%; pH at 5, 5.5, 6, 6.5, and 7) were used. The experimental design was referred to Zhu et al. [28].

### 2.5. Response Surface Optimization Design and Model Verification

Based on the results of single-factor experiments, a Plackett–Burman design was used to screen five factors to screen significant factors that affect bacterial biomass and protein expression, using Design Expert 11.0.3 (Stat-Ease Inc., Minneapolis, MN, USA). A three-factor, three-level Box–Behnken design was designed based on the response values of bacterial biomass and protein expression level. In order to validate the acquired model, the adequacy of the model fit and the assumptions for linear regression were assessed. Subsequently, the process was replicated three times under the optimized conditions.

### 2.6. Glucose Feedback Feeding and Residual Sugar Determination in Fermentation Broth

The process set parameters were inoculum size 2%; pH 6.0, supplemented with 2.5% (*w*/*v*) ammonia solution; temperature, 36.5 °C; stirring speed, 200 r/min; and dissolved oxygen, 9%. With a constant feed rate of 10 mL/h glucose solution (50% *w*/*w*), bacterial density, residual sugar, and target protein concentration were measured every 2 h over a 24 h period. Glucose solution was used for the standard curve with a concentration of 0, 0.2, 0.4, 0.6, 0.8, and 1.0 mg/mL as standard solution, after which 2 mL of the DNS reagent was added and boiled for 5 min. The reaction mixture was diluted by adding 9 mL distilled water and the absorbance was measured at 540 nm. A standard curve was established based on absorbance and glucose concentration. Fermentation broth was sampled to detect the remaining glucose content as previously described; experiments were repeated in triplicates.

### 2.7. Determination of Minimum Inhibitory Concentration (MIC_50_) by Dilution Method with Trace Meat Broth

MIC_50_ were determined as previously described, with slight modifications [29]. The active concentration of the LR-LFCA bacterial suspension was adjusted to 1 × 10^5^ CFU/mL. After centrifugation, cell-free supernatants were subjected to 50-fold trichloroacetic acid (TCA) precipitation for protein concentration. Bovine lactoferrin peptide was diluted in water to a final concentration of 768, 384, 192, 96, 48, 24, 12, 6, 3, and 1.5 mg/L and the pH was adjusted to 7.0. Chloramphenicol (10 μg/mL) was diluted using sterile PBS to a concentration of 80, 40, 20, 10, 5, 2.5, 1.25, 0.625, 0.3125, and 0.1563 μg/mL. Bovine lactoferrin peptide standards, synthesized through chemical means, were diluted using sterile PBS to achieve concentrations of 1, 0.5, 0.25, 0.125, 0.125, 0.0625, 0.03125, 0.01563, 0.00781, 0.00391, and 0.002 mg/mL. Different concentrations of LR-LFCA supernatant, bovine lactoferrin peptide standards, and chloramphenicol were added to the bacterial suspension (*S. aureus* CVCC43300 and *E. coli* CVCC10141), incubated at 37 °C for 12 h. The absorbance at 600 nm was quantified using a microplate reader. The minimum inhibitory concentration (MIC_50_) was determined as the lowest concentration of the molecule that resulted in a 50% reduction in absorbance compared to the control.

### 2.8. Oxford Cup Method for Measuring the Diameter of the Inhibition Zone

The test microorganism selected for this test was *S. aureus* ATCC43300 and *E. coli* CVCC10141. The bacterial solution was diluted to 10^6^ CFU/mL with PBS. Then, 100 μL of bacterial suspension was spread on LB plates and divided into 3 areas. Sterile Oxford cups (φ = 6 mm) were placed on each area of the plate, and then 0.2 mL of test agents was added to the Oxford cup, including (1) optimized LR-LFCA supernatant (cultured until reaching 10^9^ CFU/mL, sterile filtration and adjusted pH to 7.0); (2) unoptimized LR-LFCA supernatant (treated with the same method as (1)); and (3) sterile PBS (control group). Bacteria were cultured for 12 h at 37 °C, and then we tested the antibacterial activities by measuring the diameter of inhibitory zones. The experimental design was referred to Niamah et al. [25].

### 2.9. Animal Experiment Design

All experimental protocols were approved by the Animal Care and Use Committee of Northeast Agricultural University. The approval code was NEAUEC202303102. A total of twelve healthy ternary crossbred (Duroc × Landrace × Large White) weaned piglets, with an average initial weight of 11.03 ± 0.35 kg at 30 days of age, were randomly divided into three treatment groups with four replicates per group: (1) antibiotic-free basal diet (CON group); (2) antibiotic-free basal diet (LPS group); and (3) antibiotic-free basal diet supplemented with LR-LFCA at a dose of 1 × 10^11^ CFU/kg diet (LR-LFCA group). The experimental period lasted 21 days for each experiment, group LR-LFCA received an antibiotic-free basal diet supplemented with LR-LFCA fermentation liquor from day 1 to day 10, then fed with basal diet only. Throughout the experimental period, all piglets were provided with unrestricted access to water and feed.

The pens underwent regular disinfection and deworming in accordance with farm requirements. Experimental diets were based on corn and soybean meal, with nutritional requirements for piglets weighing 11 to 25 kg formulated based on NRC guidelines [30]. On the final day (day 21), piglets in LPS and LR-LFCA groups were administered LPS (*Escherichia coli* serotype 055: B5, Sigma Chemical Inc., St. Louis, MO, USA) intraperitoneally at doses of 100 μg/kg. Blood and tissue samples were collected 4 h post-injection.

### 2.10. Sample Collection, Processing, and Index Determination

Blood was collected from the anterior vena cava of piglets following a 4-h intraperitoneal injection of LPS or physiological saline. Blood samples were collected with EDTA anticoagulation tubes for routine blood indicators, including changes of white blood cells and red blood cells. Additionally, sterile centrifuge tubes were utilized to place whole blood, which underwent centrifugation to isolate serum. The isolated serum was subsequently stored at a temperature of −80 °C to further analysis. After euthanasia, the ileums were collected, fixed in 4% paraformaldehyde, stained with H and E and further processed for histopathological analysis. HE staining was performed as described previously [31]. Histopathological changes in the piglets from each group were observed utilizing light microscope. Intestinal tissues and intestinal mucus were harvested and stored at −80 °C. Intestinal mucus and serum LPS levels were measured with an ELISA kit (Xiamen Bioendo Technology Co., Ltd., Xiamen, China). Levels of sIgA, IgM, and IgG in intestinal mucus and serum were detected with ELISA kits (Jiangsu Meimian industrial Co., Ltd., Yancheng, China). The measurement procedures were executed in strict adherence to the guidelines provided by the product manufacturer.

### 2.11. Determination of Tight Junction Proteins and Inflammatory Factors in Piglet Intestinal Tract

Tissues (jejunum and ileum) were frozen and ground to powder in liquid nitrogen, and total RNA were extracted with the RNA Purification Kit (Sevenbio, Beijing, China). Gene expression of occludin, ZO-1, TNF-α, IL-6, IL-1β, IL-10, and TGF-β were evaluated by qPCR on ABI7500 (Thermo Fisher Scientific, Waltham, MA, USA). Gene-specific primer sequences were listed in Appendix A. Tissues (jejunum and ileum) were homogenized using a homogenizer, supernatant was collected by centrifuging at 2500 r/min for 20 min. Tight junction related protein (ZO-1 and occludin) levels in intestinal tissues were determined according to the manufacturer’s instructions (Jiangsu Meimian industrial Co., Ltd., Yancheng, China).

### 2.12. Data Processing and Statistical Analysis

Design-Expert statistical software (version 11.0.3, Stat-Ease Inc., Minneapolis, MN, USA) was used to perform the Plackett–Barman and Box–Behnken design experiments, data analysis, model establishment, and response surface analysis. One-way ANOVA analyses were performed with SPSS Statistics v. 22 (IBM SPSS Statistics, Chicago, IL, USA), and means were compared using Duncan’s multiple comparison tests. *p* values less than 0.05 were considered significant (* *p*  <  0.05; ** *p*  <  0.01). The results are presented as mean ± SD and GraphPad Prism 8.0.2.263 was applied to plot the graphs.

## 3. Results

### 3.1. Biological Characteristics of LR-LFCA and Single-Factor Optimization of Fermentation Conditions

Figure 1A depicts the growth curve for recombinant strain LR-LFCA cultured in MRS for 24 h. The results indicated that under static conditions, the lag phase lasted for about 2 h, and then LR-LFCA entered the logarithmic growth period at 2–12 h and the stationary phase at 12–24 h. The detection of LFCA expression was performed through Western blot analysis (Figure 1B), approximately 38 kDa, were detected in both the bacterial precipitate and supernatant. A quantitative standard curve for LFCA expression in LR-LFCA was established (Appendix A) using the formula y = 0.0754x + 0.0534. Quantitative ELISA assay demonstrated that the lactoferrin peptide content in the supernatant was 1.33 μg/mL and 3.24 μg/mg in the bacterial cells.

In order to investigate the impact of various factors on biomass and protein expression concentration of LR-LFCA, distinct variables were manipulated under identical culture conditions. Specifically, the effects of different inoculation amounts (Figure 1C), temperatures (Figure 1D), stirring speeds (Figure 1E), dissolved oxygen (Figure 1F), and pH values (Figure 1G) were examined. The results indicate that the highest OD_600_ value and maximum protein concentration of 8.5290 and 8.5811 mg/L, respectively, were achieved at an inoculation rate of 2%. The optimal temperature for maximum biomass and protein expression levels was found to be 37 °C, with corresponding values of 7.5533 and 8.5277 mg/L, respectively. Both the stirring speeds of 200 r/min and 400 r/min yielded elevated concentrations of biomass and protein. Nevertheless, the heightened shear force at 400 r/min may impede cell growth. Dissimilar levels of dissolved oxygen had a more pronounced impact on the expression of target protein than on cell growth, with the most elevated level of expression being observed at 10% dissolved oxygen. At pH 6.0, LFCA expression reached 9.9281 mg/L, and the OD value also attained its maximum at 7.7182.

### 3.2. Screening Significant Factors Affecting LR-LFCA Fermentation by Plackett–Burman Design

In this investigation, a Plackett–Burman (PB) experimental design was employed to evaluate major factors that influence LR-LFCA fermentation. Specifically, five factors namely temperature, pH, inoculation, dissolved oxygen level, and stirring speed, were designated as X1, X2, X3, X4, and X5, respectively, with two levels each. Experimental factors were coded as -1 for low level and 1 for high level. The PB design with N = 12 was utilized, and bacterial biomass was chosen as the response value Y1, while protein concentration (mg/L) was selected as the response value Y2. Table 1 presents the experimental design and results. Partial regression coefficients and factor significance analysis were employed to analyze the results of the PB experiment, and the outcomes of the analysis are displayed in Table 2. The models for both response indicators were found to be highly significant (*p* < 0.01), indicating the efficacy of the model. The primary and secondary factors order of influencing bacterial biomass, represented by response value Y1, were pH (X2) > dissolved oxygen (X4) > temperature (X1) > stirring speed (X5) > inoculation (X3). Similarly, the primary and secondary factors order of affecting protein concentration, represented by response value Y2, were pH (X2) > temperature (X1) > dissolved oxygen (X4) > stirring speed (X5) > inoculation (X3). Based on a comprehensive evaluation of both response indicators, pH (X2), temperature (X1), and dissolved oxygen (X4) were identified as the significant factors for subsequent response surface optimization design.

### 3.3. Optimization of LR-LFCA Fermentation Conditions by Response Surface Analysis

pH, dissolved oxygen, and temperature were coded as A, B, and C, respectively. The response value R1 and R2 were determined as bacterial biomass and protein concentration (mg/L), respectively. To optimize the fermentation conditions, the Box–Behnken design optimization method with N = 12 was employed, and each experiment was conducted in parallel. Table 3 presents the experimental design and outcomes. The multivariate regression analysis on the experimental results in Table 3 was conducted using the response surface analysis software Design-Expert.10. The polynomial regression equations for pH, dissolved oxygen, temperature, bacterial biomass (R1), and protein concentration (R2) were derived as follows:R1 = 9.18 + 0.54A − 0.17B − 0.26C − 0.075AB − 0.32AC + 0.092BC − 0.71A^2^ − 0.76B^2^ − 0.71C^2^
R2 = 17.24 − 1.00A − 0.41B + 0.16C + 0.51AB − 0.38AC − 0.33BC − 1.60A^2^ − 0.81B^2^ − 0.75C^2^

Furthermore, the variance analysis and fitting parameters of the regression model presented in Table 4 demonstrate the high significance (*p* < 0.01) of the prediction model, with a determination coefficient of R^2^ = 0.9779 (fermentation biomass) or R^2^ = 0.9788 (protein concentration), indicating a robust level of model fitting and reliability. The response surface plots depicting dissolved oxygen and pH evinced a conspicuous curvature with elliptical contours, signifying a substantial interaction between the two factors that influence bacterial growth, as illustrated in Figure 2. This observation aligns with the variance analysis presented in Table 4. Furthermore, the response surface diagrams in Figure 3 evinced a curvature in the interaction between any two of pH, temperature, and dissolved oxygen, indicating that their interaction influences recombinant protein expression. The significance of the dissolved oxygen and temperature interaction was found to be lower than that of the dissolved oxygen and pH or pH and temperature interactions.

By integrating the regression equations for the two response values, the optimal fermentation conditions for LR-LFCA were projected to be pH 6.12, temperature 36.51 °C, and dissolved oxygen 9.20%. Under these conditions, the predicted values for bacterial biomass and target protein concentration were 9.203 and 17.27 mg/L, respectively. To validate these predictions, three independent experiments were performed with adjusted fermentation parameters of 2% inoculation, pH 6, temperature 36.5 °C, dissolved oxygen 9%, and stirring speed 200 r/min. Results are expressed as OD_600_ values (9.180 ± 0.286) and protein concentration (16.94 mg/L). The relative error between the predicted and actual experimental values was found to be less than 5%. The result of glucose feedback supplementation experiment was shown in Appendix A. Two hours following fermentation, LR-LFCA entered logarithmic phase of growth, with growth rate increased rapidly. Four hours following fermentation, 50% (*w*/*w*) glucose was supplemented at a flow rate of 10 mL/h, resulted in maximum OD_600_ (10.723 ± 0.748) and a 1.17-fold surge in biomass relative to batch fermentation. Furthermore, LFCA concentration increased by 1.15-fold, reaching 19.58 mg/L.

### 3.4. The Antibacterial Activity of LR-LFCA Fermentation Supernatant after Optimization

The findings, as illustrated in Figure 4, demonstrate that chloramphenicol exhibited an MIC_50_ value of 10 μg/mL. The MIC_50_ values for *E. coli* CVCC10141 and *S. aureus* ATCC43300 in response to bovine lactoferrin peptide were 31.25 μg/mL and 15.63 μg/mL, respectively. Furthermore, the MIC_50_ values for *E. coli* CVCC10141 and *S. aureus* ATCC43300 in response to LR-LFCA supernatant were 96 μg/mL and 48 μg/mL, respectively. Although the MIC_50_ of LR-LFCA supernatant was notably higher than that of antibiotics and bovine lactoferrin peptide standards, it demonstrated a robust antibacterial effect, particularly against Gram-positive *S. aureus* ATCC43300. The antibacterial efficacy of the bovine lactoferrin peptide produced by the optimized LR-LFCA was evaluated utilizing the Oxford cup method, and the outcomes are presented in Figure 4D and Appendix A. In general, the bovine lactoferrin peptides exhibited varying levels of antibacterial activities against different pathogenic bacteria, with a particularly strong effect observed against *S. aureus* ATCC43300. The supernatant of optimized LR-LFCA resulted in a proportional increase in the diameters of bacteriostatic circles for both *E. coli* and *S. aureus*.

### 3.5. The Regulatory Effect of Optimized LR-LFCA on LPS-Induced Intestinal Inflammation in Piglets

Appendix A presented routine blood examinations. The LPS group exhibited a noteworthy reduction in mean hemoglobin content and concentration (*p* < 0.05), which was mitigated by the LR-LFCA group. No significant alterations were observed in the total red blood cell count and hematocrit across the three groups (*p* > 0.05). The LR-LFCA group significantly decreased the total white blood cell count and lymphocyte count in the LPS group (*p* < 0.05), thereby normalizing it to the CON group. The LPS group exhibited a noteworthy elevation in both neutrophil count and proportion (*p* < 0.01), which was mitigated by the LR-LFCA group. Additionally, the LP-LFCA group ameliorated the significant rise in monocyte count and platelet content that was observed in the LPS group (*p* < 0.05). Figure 5A depicts alterations in IgM and IgG levels in piglet serum, as well as sIgA content in the jejunum and ileum. The LR-LFCA group had higher total IgM and IgG levels than CON (*p* < 0.01) and LPS group. Figure 5B illustrates a significant increase in sIgA secretion level in the jejunum of the LPS group when compared to the CON group (*p* < 0.01). However, the LR-LFCA group displayed a decrease in sIgA secretion level when compared to the LPS group (*p* < 0.01) and demonstrated a trend similar to the CON group (*p* > 0.05).

This investigation analyzed the impact of LPS treatment and fermented bacterial solution of LR-LFCA on cytokine expression levels in jejunum and ileum of piglets (as depicted in Figure 5C,D). The findings revealed that LPS treatment significantly augmented the expression levels of pro-inflammatory cytokines TNF-α, IL-1β, and IL-6 in the jejunum tissue (*p* < 0.01), whereas the LR-LFCA group effectively mitigated their expression levels (*p* < 0.01). Regarding the anti-inflammatory cytokines IL-10 and TGF-β, both the LPS and LR-LFCA groups exhibited significantly lower expression levels compared to the CON group (*p* < 0.01). However, the LR-LFCA group demonstrated a significantly higher level of IL-10 than the LPS group (*p* < 0.01). In the ileum tissue, similar levels of TNF-α, IL-1β, and IL-6 were observed. However, the expression level of TGF-β in the LPS group exhibited a significant increase in comparison to the other two groups (*p* < 0.01). Conversely, the levels of TGF-β expression differed only modestly between CON and LR-LFCA group (*p* > 0.05). These results indicate that LR-LFCA possessed the capability to regulate cytokine expression levels and mitigate the upregulation of pro-inflammatory cytokines in the intestinal mucosa of piglets induced by LPS.

### 3.6. The Regulatory Effects of LR-LFCA on Intestinal Barrier Function and Intestinal Tissue Morphology in LPS-Treated Piglets

The present study aimed to assess the impact of fermented recombinant bacteria LR-LFCA on the intestinal barrier function in piglets. The findings revealed that, in comparison to the CON group, both the LPS group and the LR-LFCA group exhibited a noteworthy escalation in endotoxin concentration (*p* < 0.01); however, the LR-LFCA group significantly mitigated the rise in endotoxin content as compared to the LPS group (*p* < 0.01) and ameliorated the degree of intestinal barrier damage (Figure 6A). Both the LPS group and the LR-LFCA group exhibited a significant reduction in tight junction proteins expression in jejunum or ileum as a result of endotoxin stimulation. However, LR-LFCA demonstrated significant upregulation in mRNA expression levels of ZO-1 and occludin (*p* < 0.01) compared to the LPS group (Figure 6B). The LPS group exhibited a significant decrease in ZO-1 and occludin proteins levels in jejunum or ileum, while the LR-LFCA group exhibited the opposite trend and closed to the CON group (Figure 6C). Histological analysis revealed that the mucosal epithelial cells in the LPS group exhibited extensive necrosis and shedding, accompanied by neutrophil infiltration, mild central lacteal dilation, and mucosal edema damage (Figure 6D). In contrast, the LR-LFCA group exhibited relatively mild damage to the intestinal villi, characterized by minimal structural damage and an increase in infiltrating inflammatory and lymphoid cells. These observations suggest that LR-LFCA fermentation broth effectively mitigate the damage inflicted by LPS on the intestinal villi and enhance intestinal barrier function.

## 4. Discussion

The antimicrobial peptide, bovine lactoferrin peptide (LFCA), is derived from bovine lactoferrin and is readily absorbed. However, the current development methods (including hydrolysis and chemical synthesis) are intricate and expensive, thereby restricting their extensive usage. Consequently, genetic engineering technologies have been employed to produce large quantities of the antimicrobial peptide, with successful expression reported in diverse hosts, including Pichia pastoris X-33 [32]. The selection of appropriate host for expression of exogenous antimicrobial peptides is of paramount importance. The expression system of lactic acid bacteria has gained attention due to its stable expression level and efficient expression of target proteins [33]. Constitutive expression systems are particularly advantageous as they exhibit consistent expression levels under varying environmental conditions, thereby eliminating the need for inducers [34]. The present study reports the construction of a recombinant strain, namely pPG-LFCA-E/LR-CO21 (LR-LFCA), utilizing *LimosiLactobacillus reuteri* as a carrier for expressing LFCA, a compound that possesses both probiotic and antimicrobial peptide properties [11]. The aforementioned recombinant strain holds immense potential in enhancing piglet immunity against diseases and can be scaled up for commercial production.

The intricate process of exogenous gene expression comprises several stages, encompassing transcription, translation, protein post-translational modification, and transportation to the designated expression site [35]. The creation of recombinant bacteria establishes the capacity for the production of the desired protein, and the utilization of high-density fermentation technology is pivotal in achieving this capacity [36]. Results of recombinant bacteria fermentation are influenced by multiple factors, including pH, temperature, and dissolved oxygen, which have an impact on microbial enzyme activity and nutrient utilization efficiency [15]. The objective of this investigation was to enhance the bacterial biomass and LFCA protein expression of LR-LFCA by optimizing the culture conditions through high-density cultivation in a fermenter. With the increase in the inoculation, the biomass and target protein concentrations of the recombinant strain increased and then decreased. Additionally, the biomass and protein expression levels of the strain displayed a similar trend with increasing temperature. The highest cell density was observed at a dissolved oxygen concentration of 10–15%. Nevertheless, the LR-LFCA grew slowly and unfavorable for target protein expression under highly acidic conditions.

The optimal values for the influencing factors were determined through single-factor experiments and response surface design, resulting in inoculation with 2% seed liquid, fermentation at 36.5 °C, 9% dissolved oxygen, 200 r/min, maintenance of pH at 6 with 25% (*w*/*v*) ammonia water, and feeding with 50% glucose at 10 mL/h. Following optimization, bacterial density of OD_600_ increased by 55%, protein concentration increased by 14.92 times after 10 h of fermentation, and the growth inhibition rate of pathogenic bacteria was increased. Furthermore, it was evident that subsequent to attaining its peak value at 10 h, the optical density (OD) value experienced a substantial decline, and the bacteria underwent direct entry into the death phase, without entering the stable phase. Throughout this progression, the rate of glucose consumption was nearly negligible. This phenomenon could be attributed to the accumulation of a considerable quantity of lactic acid during the fermentation process, which impedes the growth and metabolism of the recombinant bacteria [37]. Consequently, despite the availability of adequate nutrients, the bacteria were unable to sustain their growth. These findings offer dependable process parameters for industrial fermentation production of LR-LFCA.

Due to the comparable anatomical, physiological, and immunological characteristics of the human and porcine intestine, pigs have been regarded as a desirable model for investigating the mechanisms underlying intestinal diseases and the interplay between microbes and the immune system [38,39]. Additionally, pigs exhibit a high degree of similarity to humans in terms of their immune system, with over 80% similarity, and susceptibility to pathogens [40]. Previous research has demonstrated that the expression of LFCA by recombinant *Lactococcus lactis* subsp. can enhance the growth performance and immunity of piglets [41]. *L. reuteri* has been shown to inhibit the growth of pathogenic bacteria without causing harm to the body and to colonize the gastrointestinal mucosa in a localized manner [42,43]. This study represents the initial comprehensive utilization of micro-ecological agents produced via recombinant LR-LFCA on piglets. The objective of this investigation was to assess the potential of orally administered LR-LFCA to regulate intestinal inflammation in weaned piglets.

Monitoring blood routine indicators is crucial for assessing overall health status [44,45]. Inflammation induced a marked elevation in neutrophil count in the LPS group compared to the CON group. In this context, it was noteworthy that LR-LFCA treatment ameliorated the reduction in hemoglobin levels induced by LPS, indicating a protective effect on hemoglobin and oxygen-carrying capacity [46]. Additionally, LR-LFCA demonstrated the ability to mitigate the sharp rise in sIgA in intestinal mucus induced by LPS, while maintaining stable levels of immune globulins. Overall, LR-LFCA demonstrated a favorable impact on blood routine indicators and immunity in piglets during LPS-induced inflammation.

The intestinal barrier plays a critical role in safeguarding against pathogenic invasion, and its impairment can result in intestinal dysfunction, food allergies, and irritable bowel syndrome (IBS) [47,48]. Endotoxin content is a commonly employed measure of the intestinal barrier integrity [49], and the LPS group exhibited a marked increase in endotoxin content. However, the LR-LFCA group significantly mitigated this increase, indicating that LR-LFCA can fortify the intestinal barrier integrity. The mesh structure of tight junctions between intestinal epithelial cells regulates the transport of molecules [50]. The LR-LFCA group exhibited a significant increase in the transcription and protein expression levels of occludin and ZO-1, indicating its potential to improve piglet intestinal barrier function.

The upregulation of pro-inflammatory cytokines within intestinal cells has the potential to induce functional impairment of the intestinal tissue [51,52]. Our investigation revealed a marked elevation in the expression of pro-inflammatory cytokines, namely IL-1β, L-6, and TNF-α, and a concomitant reduction in the anti-inflammatory cytokine IL-10 within the mucosa of the jejunum and ileum following LPS treatment. A recent study has demonstrated that the administration of *LimosiLactobacillus reuteri* can ameliorate intestinal inflammation by reinforcing the intestinal barrier [53]. In general, LR-LFCA had the potential to ameliorate intestinal inflammation in piglets through the augmentation of intestinal mucosal integrity, the enhancement of mucosal immunity, and other related pathways.

## 5. Conclusions

In conclusion, the optimization of the fermentation process in LR-LFCA was successful in achieving the most favorable conditions for the production of microbial biomass and protein expression. These conditions encompassed a fermentation temperature of 36.5 °C, a dissolved oxygen concentration of 9%, and a pH of 6. Furthermore, LR-LFCA fermented under these optimal conditions exhibited the ability to inhibit the growth of *Staphylococcus aureus* ATCC43300 and *Escherichia coli* CVCC10141, as well as regulate the intestinal barrier and LPS-induced intestinal inflammation. This study provides a valuable foundation for the advancement of new micro-ecological preparations and the implementation of large-scale industrial production.

## Figures and Tables

**Figure 1 foods-12-04068-f001:**
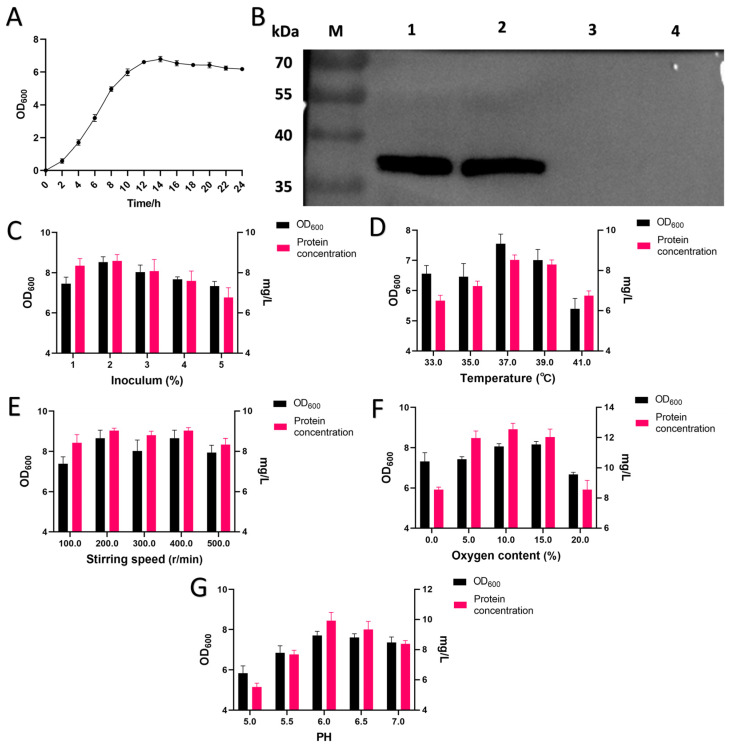
The biological characteristics of LR-LFCA and single-factor optimization of fermentation conditions. (**A**) The growth curve of LR-LFCA. (**B**) Identification of expressed LFCA by Western blot. M. Marker; 1. LR-LFCA cell precipitation; 2. LR-LFCA cell supernatant; 3. LR-CON cell precipitation; 4. LR-CON cell supernatant. (**C**) Effects of different inoculation volumes on LR-LFCA density and LFCA expression. (**D**) Effects of different culture temperatures on LR-LFCA density and LFCA expression. (**E**) Effects of different stirring speeds on LR-LFCA density and LFCA expression. (**F**) Effects of different dissolved oxygen on LR-LFCA density and LFCA expression. (**G**) Effects of different pH on LR-LFCA density and LFCA expression.

**Figure 2 foods-12-04068-f002:**
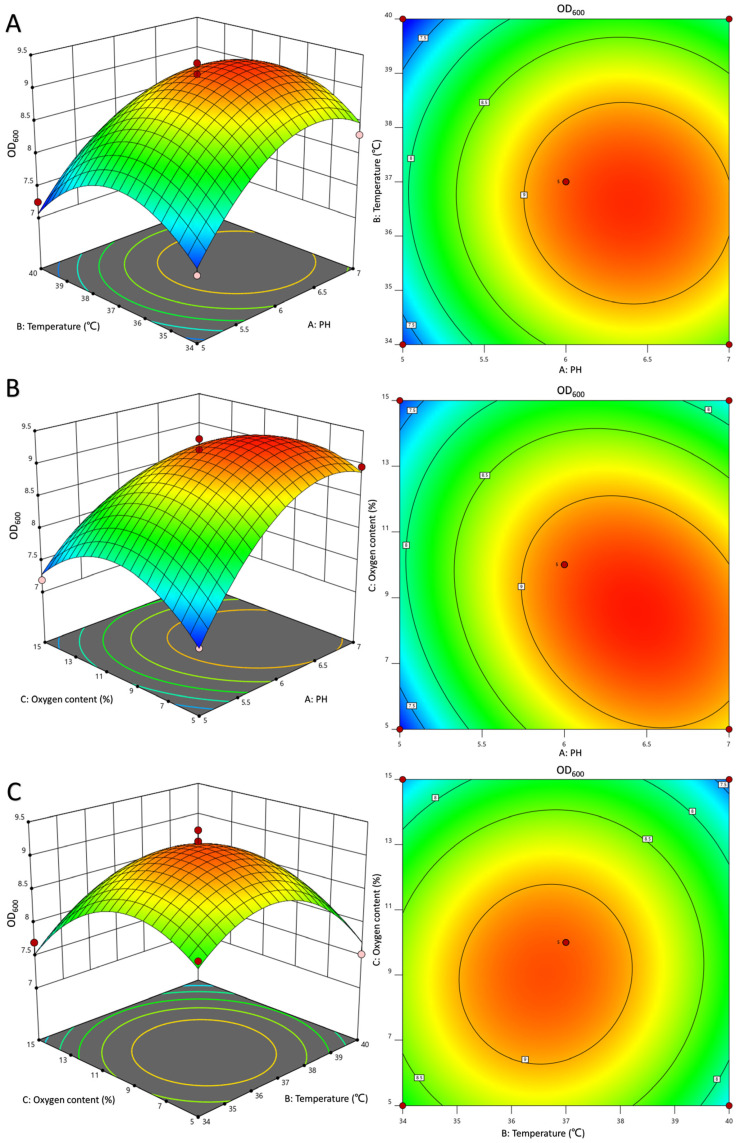
Response surface and contour map of pH, dissolved oxygen, and temperature affecting bacterial biomass. (**A**) PH and temperature. (**B**) PH and oxygen content (%). (**C**) Temperature and oxygen content (%).

**Figure 3 foods-12-04068-f003:**
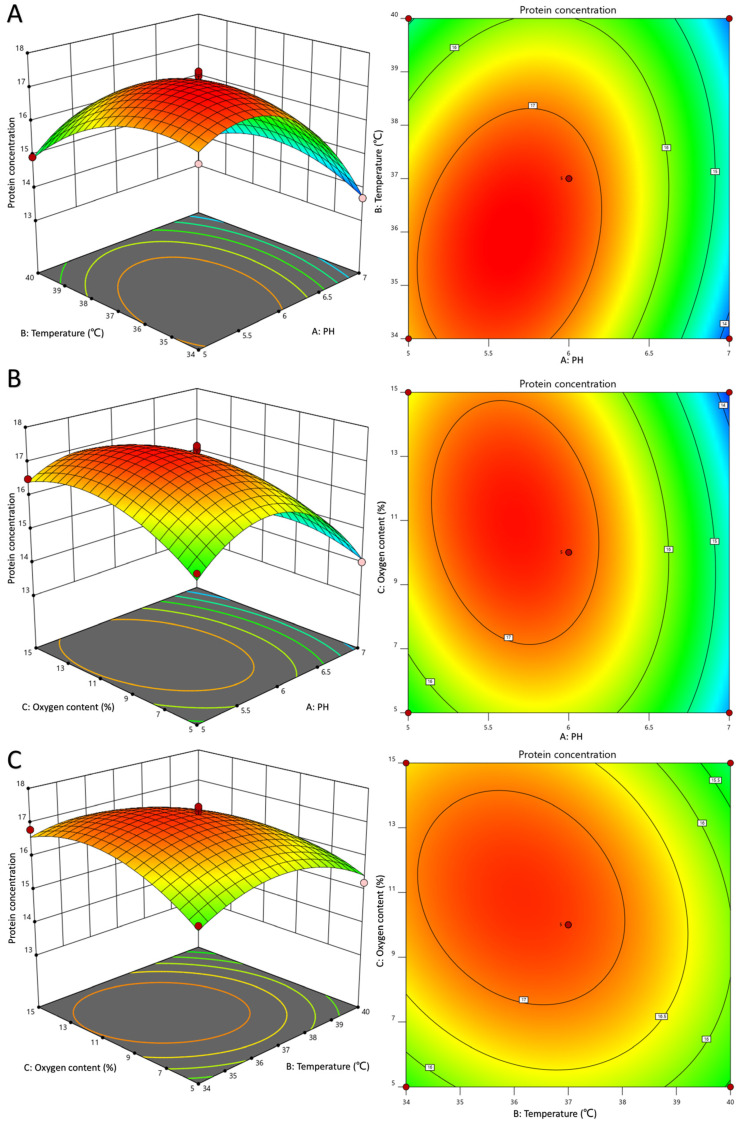
Response surface and contour map of pH, dissolved oxygen, and temperature affecting bacterial protein concentration. (**A**) PH and temperature. (**B**) PH and oxygen content (%). (**C**) Temperature and oxygen content (%).

**Figure 4 foods-12-04068-f004:**
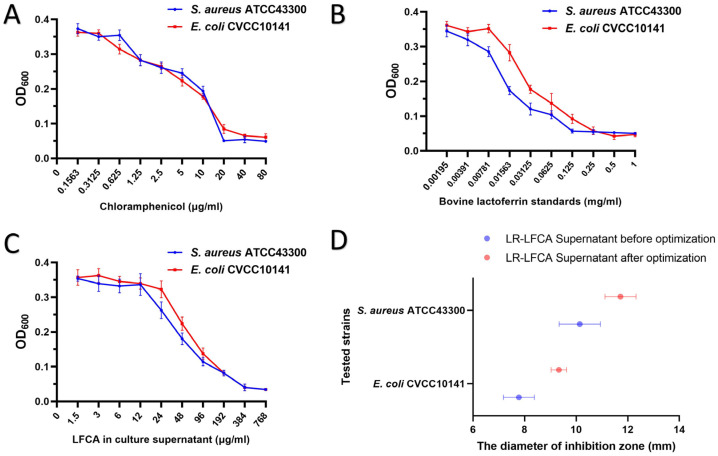
MIC_50_ and inhibition zone diameter of LR-LFCA fermentation supernatant. (**A**) Bacteriostatic curve of chloramphenicol against tested bacteria. (**B**) Bacteriostatic curve of bovine lactoferrin peptide standard against test bacteria. (**C**) Bacteriostatic curve of LFCA against test bacteria. (**D**) The diameter of inhibition zone of LR-LFCA before and after optimization.

**Figure 5 foods-12-04068-f005:**
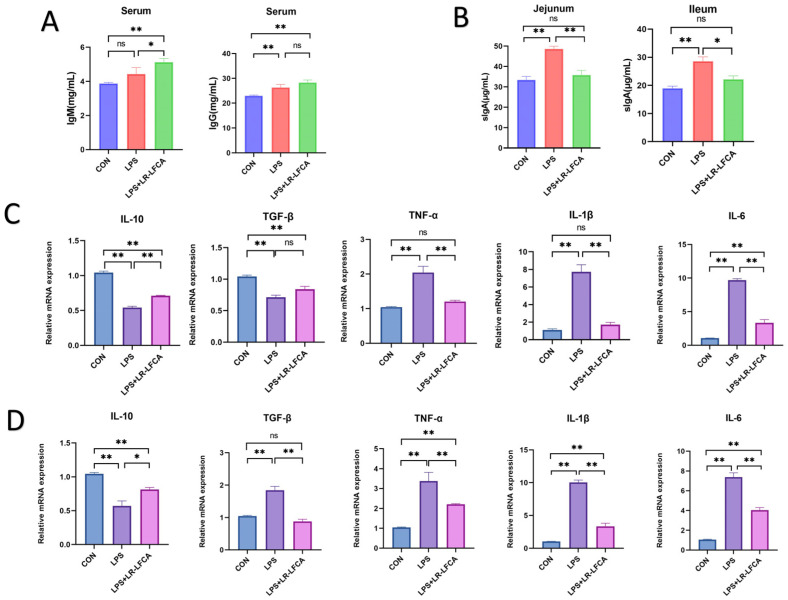
Effects of optimized LR-LFCA on serum antibody, intestinal mucosal antibody, and intestinal inflammatory factor mRNA expression of LPS-treated piglets. (**A**) Changes of IgM and IgG in serum of LPS-treated piglets. (**B**) Changes of sIgA content in intestinal mucus of LPS-treated piglets. (**C**) Changes of inflammatory factor mRNA expression in jejunum of LPS-treated piglets. (**D**) mRNA expression of inflammatory cytokines in ileum of LPS-treated piglets. ** *p* < 0.01; * *p* < 0.05; ns, not significantly different.

**Figure 6 foods-12-04068-f006:**
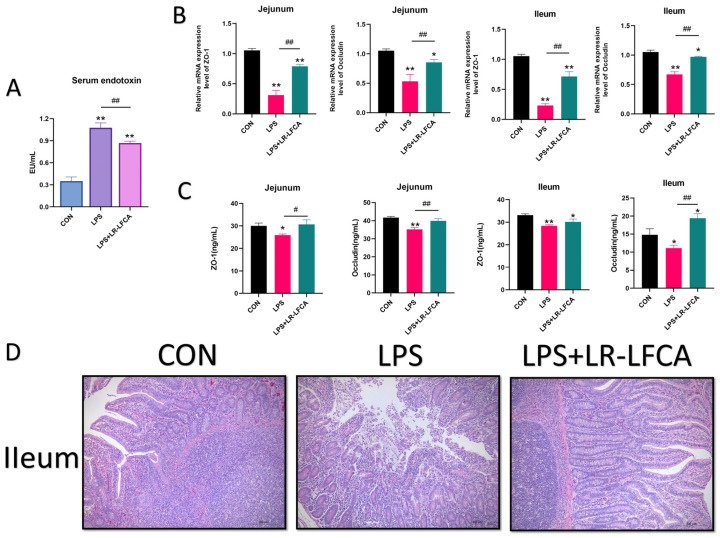
Effects of optimized LR-LFCA on intestinal barrier function and intestinal tissue morphology of LPS-treated piglets. (**A**) Changes of endotoxin in serum of piglets. (**B**) Relative mRNA expression of ZO-1 and occludin in jejunum and ileum of piglets were detected by real-time PCR. (**C**) The proteins expression levels of ZO-1 and occludin in jejunum and ileum of piglets were detected by ELISA kit. (**D**) Pathological changes and pathological sections of intestine (100×). * *p* < 0.05; ** *p* < 0.01 vs. CON; # *p* < 0.05; ## *p* < 0.01 vs. LPS + LR-LFCA.

**Table 1 foods-12-04068-t001:** Plackett–Burman experimental design and results.

Run	Temperature (°C)—X1	PH—X2	Inoculum (%)—X3	Oxygen Content (%)—X4	Stirring Speed (r/min)—X5	Biomass (OD_600_)—Y1	Protein Concentration (mg/L)—Y2
1	39	7	3	5	100	7.0081	12.224
2	35	5	1	5	100	5.3086	7.296
3	39	7	1	5	100	6.7272	11.942
4	39	7	1	15	300	8.9154	14.769
5	39	5	1	5	300	5.2155	9.927
6	39	5	3	15	300	6.0287	10.951
7	35	5	3	5	300	5.8037	7.884
8	35	5	1	15	100	6.0858	9.516
9	35	7	1	15	300	8.7379	11.984
10	35	7	3	5	300	8.0638	10.748
11	35	7	3	15	100	9.5285	11.804
12	39	5	3	15	100	5.9826	10.968

**Table 2 foods-12-04068-t002:** Significance analysis of PB results with biomass or protein concentration as response value.

Source (Biomass)	Sum of Squares	Degree of Freedom	Mean Square	F-Value	*p*-Value	Remark
Model	22.38	5	4.48	38.04	0.0002	significant
X1	0.94	1	0.94	7.95	0.0304	*
X2	16.94	1	16.94	143.93	<0.0001	**
X3	0.11	1	0.11	0.90	0.3803	
X4	3.91	1	3.91	33.25	0.0012	**
X5	0.49	1	0.49	4.16	0.0874	
Pure error	0.71	6	0.12			
Total	23.09	11				
**Source (protein concentration)**	**Sum of Squares**	**Degree of Freedom**	**Mean Square**	**F-Value**	***p*-Value**	**Remark**
Model	43.87	5	8.77	61.47	<0.0001	significant
X1	11.11	1	11.11	77.88	0.0001	**
X2	23.88	1	23.88	167.33	<0.0001	**
X3	0.061	1	0.061	0.43	0.5378	
X4	8.29	1	8.29	58.05	0.0003	**
X5	0.53	1	0.53	3.69	0.1032	
Pure error	0.86	6	0.14			
Total	44.73	11				

Note: ** *p*< 0.01, * *p*< 0.05.

**Table 3 foods-12-04068-t003:** Box–Behnken experimental design and results.

Run	PH—A	Temperature (°C)—B	Oxygen Content (%)—C	Biomass (OD_600_)—R1	Protein Concentration (mg/L)—R2
1	6	34	5	8.335	15.732
2	7	34	10	8.3025	13.696
3	5	37	15	7.1925	16.506
4	7	40	10	8.0911	14.238
5	6	34	15	7.7075	16.801
6	5	35	10	7.1625	16.448
7	6	37	10	9.3807	17.05
8	6	40	15	7.2675	14.968
9	6	37	10	9.0942	16.923
10	6	40	5	7.5275	15.227
11	6	37	10	9.0732	17.128
12	7	37	15	7.7175	13.49
13	5	37	5	7.1702	15.529
14	5	40	10	7.251	14.946
15	7	37	5	8.9599	14.023
16	6	37	10	9.2178	16.936
17	6	37	10	9.1236	17.172

**Table 4 foods-12-04068-t004:** Variance analysis of R1 or R2 response surface quadratic model.

Source (R1)	Sum of Squares	Degree of Freedom	Mean Square	F-Value	*p*-Value	Remark
Model	11.02	9	1.22	34.36	<0.0001	significant
A	2.31	1	2.31	64.68	<0.0001	**
B	0.23	1	0.23	6.58	0.0372	*
C	0.56	1	0.56	15.58	0.0056	**
AB	0.022	1	0.02	0.63	0.4532	
AC	0.4	1	0.40	11.22	0.0123	*
BC	0.034	1	0.03	0.95	0.3629	
A^2^	2.14	1	2.14	60	0.0001	
B^2^	2.45	1	2.45	68.83	<0.0001	
C^2^	2.09	1	2.09	58.73	0.0001	
Residual	0.25	7	0.036			
Lack of Fit	0.19	3	0.062	3.9	0.111	not significant
Pure Error	0.064	4	0.016			
Total	11	16				
R^2^ = 0.9779	(C.V.%) = 2.32					
**Source (R2)**	**Sum of Squares**	**Degree of Freedom**	**Mean Square**	**F-Value**	***p*-Value**	**Remark**
Model	29.07	9	3.23	35.89	<0.0001	significant
A	7.96	1	7.96	88.48	<0.0001	**
B	1.36	1	1.36	15.11	0.006	**
C	0.2	1	0.2	2.18	0.183	
AB	1.04	1	1.04	11.6	0.0113	*
AC	0.57	1	0.57	6.33	0.04	*
BC	0.44	1	0.44	4.9	0.0625	
A^2^	10.81	1	10.81	120.12	<0.0001	
B^2^	2.74	1	2.74	30.5	0.0009	
C^2^	2.38	1	2.38	26.48	0.0013	
Residual	0.63	7	0.09			
Lack of Fit	0.4	3	0.13	2.24	0.2257	not significant
Pure Error	0.23	4	0.059			
Total	30	16				
R^2^ = 0.9788	(C.V.%) = 1.90					

Note: ** *p* < 0.01, * *p* < 0.05.

## Data Availability

Data are contained within the article.

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
