# Peer review of "Optimum Fermentation Conditions for Bovine Lactoferricin-Lactoferrampin-Encoding LimosiLactobacillus reuteri and Regulation of Intestinal Inflammation"

_foods, 2023, doi:10.3390/foods12224068_

Round 1

Reviewer 1 Report

Comments and Suggestions for Authors

The manuscript entitled Optimum fermentation conditions for bovine lactoferricin-lactoferrampin-encoding Lactobacillus reuteri and regulation of intestinal inflammation presents information related to the optimization process for the production of lactoferricin and lactoferrampin by L. reuteri and the evaluation as anti-inflammatory agents. The manuscript presents several issues that the authors must attend to. Below are the comments.

- Line 36. Define Lfcin.

-Line 39. Define LFampin.

-Line 116. What was the rationale for using 3 L of fermentation medium into a 12 L fermentor? Was the airspace into the fermentor important for microbial growth?

-Sections 2.4 and 2.5. What was the rationale for screening factors using single-factor evaluation? In section 2.5 the authors used the Plackett-Burmann exploratory design. Why not use only Plackett-Burman? The design allows the evaluation of several factors at the time.

-Line 263. How did the authors determine the lag phase, log phase, and stationary phase duration? Static conditions? The authors mentioned in the methodology that they used 100 rpm.

-Figure 1. Improve the presentation of the figure. The data cannot be seen clearly. Split the figure into A-D and D-H. What was the rationale for showing Fig. 1C? The figure must be placed as supplementary material 

Comments on the Quality of English Language

The English is fine. Minor revision is required.

Reviewer 2 Report

Comments and Suggestions for Authors

Dear editors and authors 

The article is well conceived. It contains all the necessary chapters. The introduction is written correctly and provides enough data to understand the goal of the research but it needs some corrects in methods and results chapter.

1-Authors must use the modern nomenclature of lactic acid bacteria throughout the manuscript and correct scientific names, such as Lactobacillus reuteri correct to Limosilactobacillus reuteri .

2-There is no clear goal in the manuscript. A goal must be added at the end of the introduction and be clear and clear to the reader. 

3-There are many work methods without scientific references, and it is difficult for the reader to go back and verify the work method used. I suggest you add these references.

Niamah, A. K., Mohammed, A. A., & Alhelf , N. A. (2023). ANTIBACTERIAL ACTIVITY AND IDENTIFICATION OF PRODUCED REUTERIN FROM LOCAL Lactobacillus reuteri LBIQ1 ISOLATE. Journal of Microbiology, Biotechnology and Food Sciences, 12(5), e4701. https://doi.org/10.55251/jmbfs.470.

-Saroha, T., Sharma, S., Choksket, S., Korpole, S., & Patil, P. B. (2023). Limosilactobacillus walteri sp. nov., a novel probiotic antimicrobial lipopeptide-producing bacterium. FEMS Microbiology Letters, 370, fnad004.

4-Lines 104-109: A reference should be added to this paragraph.

5-Lines 110-126: A reference should be added to this paragraph.

6-Lines 129-146: A reference should be added to this paragraph.

7-Lines 148-158: A reference should be added to this paragraph.

9-The volume of bacterial inoculum added is 1-5. This volume does not mean anything. You must remember the number of viable organisms in this volume of the bacterial inoculum.

10-Lines 179-194: A reference should be added to this paragraph.

11-Western blot analysis (Figure 1B), approximately 38 kDa, Figure 1b does not show the molecular weight of the band at 38 kDa and the weight appears less than that.

12-Pictures of the targeted bacterial inhibition must be added to the Supplementary file.

13-The conclusions must be rewritten correctly. What is written in the manuscript are not conclusions, but rather results. In this chapter, we do not write any conclusion, but rather we mention the conclusion from the research or study.

Reviewer 3 Report

Comments and Suggestions for Authors

The study is quite interesting, the tables and the figures summarized the main findings very well, following are some comments.

-Add more numerical findings in the abstract (the significance level).

L22: add the complete meaning of 200 r/min.

L 191: add the model and details of incubator, as well for all instrumenst used.

Improve the conclusion part by add the main insights, limitations and the future perspectives.

Comments on the Quality of English Language

Can be improved

Round 2

Reviewer 2 Report

Comments and Suggestions for Authors

Dear Editors,

I now advise publishing the paper in its current version because the authors made all the required improvements.